# Predicting Factors for Pancreatic Malignancy with Computed Tomography and Endoscopic Ultrasonography in Chronic Pancreatitis

**DOI:** 10.3390/diagnostics12041004

**Published:** 2022-04-15

**Authors:** Jian-Han Lai, Keng-Han Lee, Chen-Wang Chang, Ming-Jen Chen, Ching-Chung Lin

**Affiliations:** 1Division of Gastroenterology, Department of Internal Medicine, MacKay Memorial Hospital, Taipei 10449, Taiwan; jiannhann@gmail.com (J.-H.L.); gun60224@gmail.com (K.-H.L.); mky378@yahoo.com.tw (C.-W.C.); mingjen.ch@msa.hinet.net (M.-J.C.); 2Department of Nursing, MacKay Medicine, Nursing and Management College, Taipei 11260, Taiwan; 3Department of Medicine, MacKay Medical College, New Taipei 25245, Taiwan

**Keywords:** endoscopic ultrasonography, computed tomography, chronic pancreatitis, pancreatic ductal adenocarcinoma

## Abstract

Diagnosing pancreatic malignancy is challenging, especially in patients with chronic pancreatitis (CP). Endoscopic ultrasonography (EUS) is a promising diagnostic procedure for discriminating between malignancy and CP. We aimed to investigate the predictive factors and reliability of computed tomography (CT) and EUS for differentiating pancreatic mass lesions and the diagnostic accuracy of EUS-FNA or FNB in patients with CP. Forty patients with CP, receiving CT and EUS-FNA or FNB for pancreatic mass lesion evaluation, were enrolled in the study. Patients’ data, CT and EUS characteristics, image-based diagnosis, cytopathology, and final diagnosis were recorded. EUS was superior to CT in terms of diagnostic accuracy (92.5% vs. 82.5%, *p* = 0.02). Both CT and EUS showed significant predictive factors (all *p* < 0.05) with the tumor image hypoattenuation pattern or vessel invasion on CT and pancreatic duct dilatation, or distal pancreatic atrophy on EUS. EUS imaging is a reliable modality for evaluating pancreatic lesions, even with a CP background. The EUS image has a higher diagnostic accuracy than CT. Predicting factors, including hypoechoic pattern, pancreatic duct dilatation, and distal pancreas atrophy, may help to differentiate benign or malignant in patients with CP.

## 1. Introduction

Pancreatic ductal adenocarcinoma (PDAC) is a highly aggressive cancer that accounts for 466,000 deaths annually worldwide. It is the seventh leading cause of cancer-related deaths despite its relatively low but increasing incidence [1]. Although diagnostic modalities have improved, it remains challenging to discriminate between PDAC and chronic pancreatitis (CP). CP can present as a pseudotumor [2] caused by chronic inflammation and is a major risk factor for PDAC [3]. While Whipple procedure, pancreaticoduodenectomy with curative aim, offers an improved survival rate for patients with resectable disease (mortality rate <5%), the complications of the Whipple procedure can be as high as 40–50%, producing a markedly worse quality of life [4].

Differentiating between PDAC and CP before an operation is important for subsequent treatment and prognosis, thus developing modalities that are less invasive and have higher diagnostic yield are imperative.

Computed tomography (CT) is an important tool that has greatly improved our ability to depict pancreatic masses. It is the standard diagnostic procedure for PDAC. However, it is not always useful, especially in patients with CP, and discrimination between focal pancreatitis and pancreatic cancer is a well-known dilemma [5]. Endoscopic ultrasonography (EUS) is an ultrasound technique in which the tip of the endoscope is equipped with a high-frequency transducer. High-resolution images of the pancreas can be obtained through the gastrointestinal tract without the disrupting effects of intervening bowel gas, providing more detailed evaluation of both solid [6] and cystic [7] pancreas lesions. While contrast-enhanced-EUS (CE-EUS) was viewed as a promising tool in discriminating pancreas lesions ever since its first publication in 1995 [8,9], the availability of the microbubble-containing contrast medium hindering the universal application of CE-EUS, making conventional EUS still the examination most frequently conducted, and the superiority of conventional EUS and CT in investigating pancreas neoplasm has yet to reach consensus [10,11].

Besides image study, biopsy is the most reliable diagnostic method for differentiating mass-like lesions in the pancreas. Both EUS-guided fine needle aspiration (FNA) [12] and fine needle biopsy (FNB) [13] are validated diagnostic procedure in obtaining pancreas specimen for further pathology study. The complication rate is low, approximately 0–2.5% in previous studies [14]. Although rare, lethal complication following EUS-FNA/FNB could happen, which is generally related to fetal hemorrhage and progression of bowel perforation [15]. Moreover, sampling error leading to false-negative cytopathology results is also noteworthy [16]. Thus, multiple studies have demonstrated that EUS plays an important role in the clinical evaluation of pancreatic solid tumors.

The aim of this study was to investigate the predictive factors and reliability of the two most common imaging modalities (CT and EUS) for differentiating pancreatic mass lesions, and the diagnostic accuracy of EUS-FNA or FNB in patients with CP.

## 2. Materials and Methods

### 2.1. Patients

A total of 276 adult patients underwent EUS-FNA/FNB for pancreatic tumors at MacKay Memorial Hospital between January 2017 and April 2021. Among these patients, 42 with CP were considered eligible for inclusion. The diagnosis of background CP was made using EUS based on the proposed Rosemont criteria (Figure 1). Diagnosis and suggestive CP were defined as CP in this study [17]. We excluded one patient in whom the final diagnosis was uncertain, and one patient did not undergo CT examination. Patients who were diagnosed with benign lesions underwent image follow-up for at least 6 months to rule out the possibility of a missed malignancy diagnosis. Finally, our analysis was based on the data of 40 patients with a confirmed diagnosis, including 23 patients with malignancy and 17 patients with benign lesions.

### 2.2. Study Design

This was a retrospective study designed to evaluate the accuracy of EUS and CT scans in diagnosing pancreatic tumors in CP. The CT and EUS images were reviewed by two gastroenterologists who were blinded to the final pathological results. The criteria of imaging for the differential diagnosis between inflammation and malignant tumors were based on a previous study [18] and the final diagnosis was made based on the cytopathological results obtained by EUS-FNA/FNB. If the clinical and imaging studies of a patient did not coincide with the FNA/FNB cytopathological diagnosis, further surgical biopsy, transabdominal echo-guided metastatic lesion biopsy or surgical resection to verify the diagnosis. Patient data were retrospectively analyzed using the hospital-patient information system. Data such as age, sex, etiology of CP, tumor characteristics on EUS and CT, FNA/FNB pass number, cytopathology results, final diagnosis, and staging of malignancy were recorded.

### 2.3. EUS-FNA/FNB Procedure Performance

All EUS-FNA/FNB procedures were performed with patients in the left lateral decubitus position under conscious sedation using midazolam (5 mg) and fentanyl (0.1 mg). Additional sedatives were administered by endoscopists as needed to achieve moderate conscious sedation. All EUS-FNA/FNB procedures were performed by two endoscopists who achieved the FNA learning curve [19]. EUS procedures were performed using a curvilinear echoendoscope (GF-UCT260, Olympus Co., Tokyo, Japan), and aspiration was performed using a 22-gauge needle (EZ Shot 3 plus aspiration needles, Olympus, Japan; Acquire^TM^, Boston Scientific Co., Natick, MA, USA). A fanning method was used, with aspiration from at least 4 different areas within the target lesion, with negative pressure applied using a 10 mL syringe. The endoscopists then fixed the acquired tissues in ethanol and formalin for preparation as cytological smears and pathological samples, respectively. Rapid on-site cytological evaluation was not performed. The endoscopists individually decided on the number of FNA/FNB passes required for each case based on the volume of obtained tissue (macroscopic on-site quality evaluation).

### 2.4. Cytopathological Analysis

The diagnostic criteria used for cytopathological diagnoses were determined by a cytopathologist. Negative means the ductal epithelium with a well-organized honey-combed pattern, uniformly sized nuclei, fine granular chromatin, and inconspicuous nucleoli. Malignant ductal epithelium includes ductal cells that have lost the honey-combed arrangement, have varying nuclear size, irregularity in nuclear contour, vesicular nuclei, and a prominent nucleolus. Based on the cytological results, patients were subdivided into the following four diagnostic groups: benign, atypical, suspicious, and positive for malignancy [20]. A “false-negative” FNA/FNB cytological diagnosis was defined as either a benign or atypical result in patients with malignancy, with a suspicious or positive finding of malignancy defined as a “positive” FNA/FNB diagnosis.

### 2.5. Statistical Analyses

Continuous variables are reported as mean ± standard deviation, and categorical variables are reported as frequencies and percentages. Independent sample *t*-test, Chi-square test, and crosstabs statistics were used, according to the data type, to compare the tumor characteristics between the malignancy and benign tumor groups. All of the analyses were performed using the SPSS software (version 21.0; SPSS, Chicago, IL, USA), with a two-sided *p*-value of 0.05, considered significant.

## 3. Results

A total of 40 patients were enrolled in this study (Figure 2). The mean age of the patients was 57 ± 15 years. Among the patients, 33 were male (82.5%). The tumors were most frequently found at the pancreatic head (24/40, 60%), and the mean tumor size was 3.09 ± 1.33 cm. All of the patients had received either FNA or FNB for cytopathology diagnosis, and seven (17.5%) subsequently underwent surgery. The final cytopathological examination revealed that 23 (57.5%) patients had pancreatic ductal adenocarcinoma. Among those who were diagnosed with malignancy, 2 (8.7%) had stage I disease, 2 (8.7%) had stage IIA disease, 3 (13.0%) had stage IIB disease, 5 (21.7%) had stage III disease, and 11 (47.8%) had stage IV disease.

The characteristics of the tumors on CT and EUS are shown in Table 1. A hypoattenuation pattern on CT was found in 18 patients (45%) and isoattenuation pattern in 22 patients (55%); hypoechoic and isoechoic patterns on EUS were found in 33 (82.5%) and seven (17.5%) cases, respectively. Pancreatic duct dilatation was recognized in 18 patients by CT and in 20 patients by EUS; distal pancreas atrophy was found in 13 patients by CT and 14 patients by EUS; vessel invasion was identified in 13 patients on CT and 11 patients on EUS. In 20 patients with pancreatic duct dilatation, most (*n* = 19) had whole upstream dilatation, and only 1 patient had focal segmental dilatation. Overall, EUS provided significantly better diagnostic accuracy than CT, as EUS was used to correctly diagnose 37 (92.5%) patients, whereas CT was used to correctly diagnose 33 (82.5%) patients (*p* = 0.02).

The frequency of the aforementioned characteristics found in pancreatic malignancy and benign tumors were further evaluated and we found that the hypoattenuation pattern and vessel invasion on CT and pancreatic duct dilatation and distal pancreas atrophy on EUS, helped differentiate PDAC from benign tumors, and were statistically significant (Table 2) (Figure 3 and Figure 4). While the difference was insignificant, the data also suggested that pancreatic malignancy was more frequently hypoechoic on EUS. CBD dilatation, PD stones, and pseudocysts were not helpful in discriminating malignancy in our study (data not shown).

Using the EUS predictive factors for differential diagnosis between malignant and benign tumors, such as hypoechoic pattern, pancreatic duct dilatation, and distal pancreas atrophy (Table 3), we stratified the tumors by the number of predicted factors that were met and found that the probability of malignancy increased with the number of predictive factors. We found that all malignant tumors had at least one of the characteristics listed above but with poor specificity, and in tumors that met three predictive factors, only one was diagnosed as a benign tumor by histopathology but with poor sensitivity.

Of the 23 patients proven to have pancreatic malignancy, 14 were diagnosed by either FNA or FNB, and nine had initial negative results of cytopathological examinations by FNA/FNB and were subsequently diagnosed with malignancy by further surgical resection (*n* = 4), surgical biopsy (*n* = 3), and transabdominal echo-guided hepatic metastatic lesion biopsy (*n* = 2). In a comparison of successful (positive) and failed (false-negative) FNA or FNB cytopathological diagnosis groups, there was no statistically significant difference in FNA/FNB, pass number, tumor location, or pancreatic parenchyma calcification. However, the smaller tumor seems to have a higher failure rate than larger tumors in patients with CP (Table 4)

We made a diagnostic pathway suggestion for the management of CP patients who had pancreatic mass based on the results of this study (Figure 5).

## 4. Discussion

In our study, EUS had a better image differential diagnosis rate (92.5%) than CT (82.5%). The reason might be that EUS was more sensitive in pancreatic duct dilatation detection, which is an important feature in pancreatic cancer diagnosis. In previous studies of the overall group, pancreatic carcinoma was more prone to hypoattenuation of the surrounding pancreatic parenchyma on CT, with some studies reporting up to 89% [10]. In our study focusing on tumor discrimination in CP, a hypoattenuation pattern was found in only 65.2% of the malignancy group, which was lower than that in previous studies. It might be a difference between CP and overall patients, and it could be related to different backgrounds. CP has a fibrotic background, which presented a relatively low attenuating image compared with normal parenchyma, and the target lesion seemed to have an iso-attenuating pattern compared with the fibrotic background.

Some secondary signs are very helpful in making a diagnosis, including pancreatic duct interruption, pancreatic duct or common bile duct dilatation, atrophic distal pancreas, mass effect, and/or convex contour abnormalities. However, the same secondary signs can also occur in mass-forming chronic pancreatitis. Some of these secondary signs by image were recorded in our study, and the most relevant features suggesting malignancy were pancreatic duct dilatation and distal pancreatic atrophy. We defined the risk factors of malignancy in EUS as a hypoechoic pattern and the above two features. If tumors meet at least two of three predictive factors, it seems to achieve acceptable sensitivity and specificity for differential diagnosis between benign and malignant tumors. However, 76.4% (13/17) of the benign tumors had at least 1 of the above characteristics, and 1 benign tumor had all the characteristics that made the image diagnosis more challenging.

Hence, some studies discussed about trying to resolve this difficult problem by contrast and/or FNA/FNB using. Contrast-enhanced EUS depicts parenchymal perfusion and provides information on the microvasculature [21]. In contrast-enhanced EUS, pancreatic carcinoma usually presents as a hypo-enhancing lesion compared to the adjacent parenchyma, while mass-forming chronic pancreatitis is expected to be iso-enhancing. It was reported the sensitivity and specificity of CE-EUS in detecting pancreas cancer could reach 93–94% and 88–89%, respectively [22,23]. Therefore, contrast enhancement may improve diagnostic yield and overall accuracy.

Besides image interpretation, EUS-FNA/FNB is thought to be a reliable tool for the evaluation of pancreatic masses. In our study, FNA/FNB correctly identified 14 out of the 23 malignancies, with a sensitivity of 60.9%. The overall diagnostic accuracy was 77.5%. This is comparable with previous reports of lower sensitivity in patients with CP (53.5%) than in patients without CP (89.3%) [24,25]. This is likely due to sampling errors in the CP fibrosis background. None of the positive FNA/FNB results were later diagnosed as benign, with a specificity of 100%, thus, no patients with benign tumors were operated on unnecessarily. Further analysis of these nine false-negative patients revealed that seven patients had suspected malignancy as either EUS or CT images. Diagnosis made with a combination of the image and FNA/FNB histopathological examination may help increase the sensitivity.

The superiority of FNA and FNB has long been debated and, in our study, while FNB sampling could achieve a higher success cytopathological diagnostic rate (85.7%, 6/7) compared with FNA (50%, 8/18) in patients with CP, the difference was not statistically significant. FNA/FNB pass number, tumor location, and parenchyma calcification were not significant factors that could influence the results. Tumor size was a significant factor in failed cytopathological diagnosis. This finding differs from the results of the overall patients [19].

Due to MRI being a relatively expensive and uncommon examination compared with CT and EUS, we did not enroll it in our study. MRI was also an important examination for differential pancreatic malignancy from benign tumors. A study found that dynamic contrast-enhanced MRI and diffusion-weighted imaging (DWI) yielded a good sensitivity, specificity, and accuracy (96.9%, 94.4%, and 96.0%, respectively) [26]. However, another study concluded that DWI could not help, and even definitive preoperative diagnosis may not be possible in some cases, especially those patients who had no multiple signs of cancer [27]. Hence, even if we had multi-modality imaging to obtain a reliable diagnosis in most CP patients, we still need to remain vigilant in order not to miss a diagnosis of cancer [28]. After all, CP patients had an increased risk of pancreatic cancer.

The limitations of our study include the following:

(1)There was no contrast enhancement of the EUS images.

By combining CE-EUS image with the predictive features that we have listed in this study, we could perhaps further enhance the accuracy of EUS and subsequently the diagnostic yields of FNA/FNB.

(2)This is a single center, small retrospective cohort study.

The quality of EUS image and the diagnostic yield of EUS-FNA/FNB is highly operator-dependent. The EUS imaging and FNA/FNB procedures of this study were completed by two experienced operators, while we had two gastroenterologist who were blinded to the final diagnosis to review the images, confounding may still exist. Further studies with larger sample size are warranted before applying the result of our study universally.

In conclusion, EUS was more sensitive to the differential diagnosis of malignant and benign tumors in patients with CP compared with CT images. The predictive factors of EUS images were hypoechoic pattern, pancreatic duct dilatation, and distal pancreatic atrophy. Tumor size may influence the FNA/FNB cytopathological diagnosis rate.

## Figures and Tables

**Figure 1 diagnostics-12-01004-f001:**
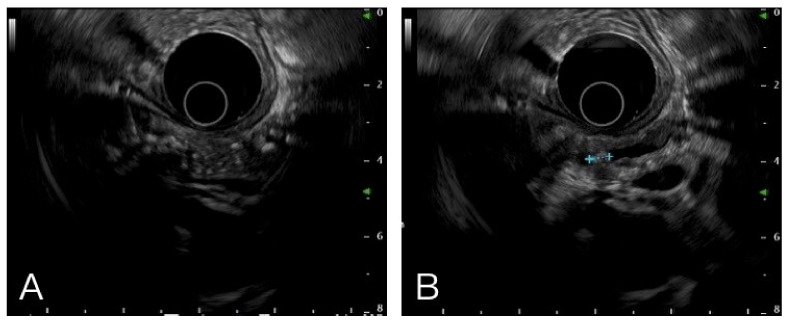
Multiple hyperechoic foci with posterior acoustic shadow (**A**) and stone in pancreatic duct (**B**) were the typical images in chronic pancreatitis.

**Figure 2 diagnostics-12-01004-f002:**
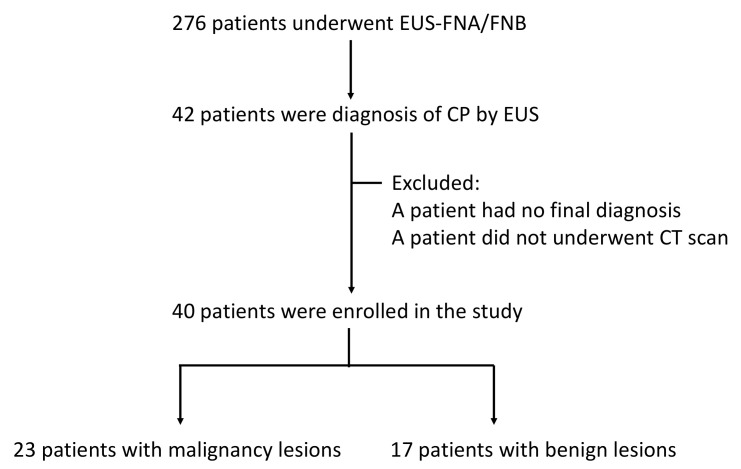
The flowchart of this study.

**Figure 3 diagnostics-12-01004-f003:**
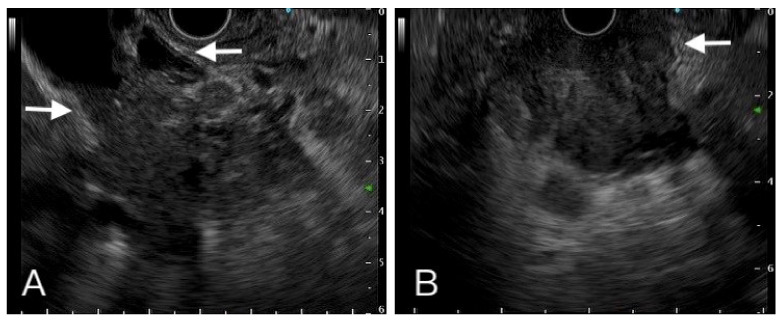
A hypoechoic heterogeneous mass had ill-defined margin with vessel (left side arrow), common bile duct (right side arrow) (**A**). Another one hypoechoic mass invaded to duodenum (**B**). Tissue proof by FNB showed these two tumors were adenocarcinoma.

**Figure 4 diagnostics-12-01004-f004:**
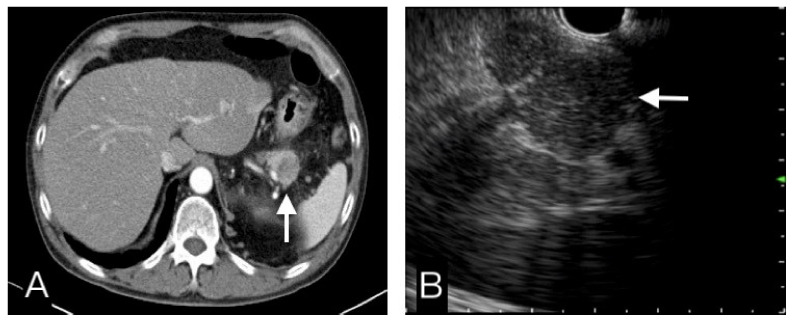
This pancreatic tail tumor had a hypo-attenuating pattern in CT (**A**), and an iso-echoic pattern in EUS (**B**). Benign etiology was more favor diagnosis than malignancy. FNB cytopathological results, clinical and image follow-up confirmed this was a benign tumor.

**Figure 5 diagnostics-12-01004-f005:**
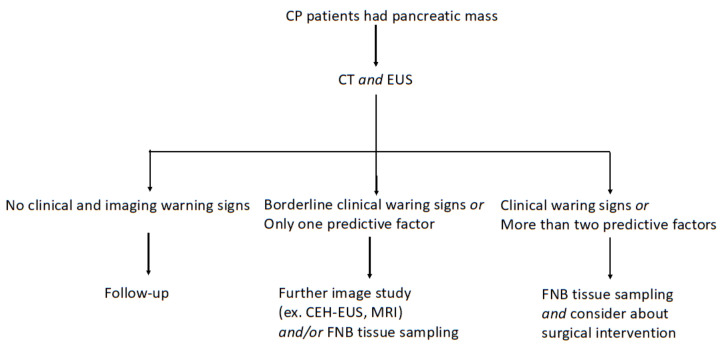
Diagnostic pathway in CP patients who had pancreatic mass.

**Table 1 diagnostics-12-01004-t001:** Tumor characters of CT and EUS image in 40 chronic pancreatitis patients.

	CT Image	EUS Image	*p*-Value
Hypo-/Iso-pattern in CT or EUS	18/22	33/7	
Pancreatic dilatation, *n* (%)	18 (45.0%)	20 (50.0%)	
Distal pancreas atrophy, *n* (%)	13 (32.5%)	14 (35.0%)	
Vessel invasion, *n* (%)	13 (32.5%)	11 (27.5%)	
Correct image diagnosis rate, *n* (%)	33 (82.5%)	37 (92.5%)	0.02

**Table 2 diagnostics-12-01004-t002:** Tumor characters to predict malignant tumors.

	Malignancy (*n* = 23)	Benign (*n* = 17)	*p*-Value
Hypo/Iso-attenuating pattern in CT	15/8	3/14	0.003
Hypo/Iso-echoic pattern in EUS	21/2	12/5	0.088
Pancreatic dilatation in EUS, *n* (%)	15 (65.2%)	5 (29.4%)	0.025
Distal pancreas atrophy in EUS, *n* (%)	12 (52.2%)	2 (11.8%)	0.008
Vessel invasion in CT, *n* (%)	12 (52.2%)	1 (5.9%)	0.002

**Table 3 diagnostics-12-01004-t003:** The number of EUS predict factors.

Predict Factors *	Benign (*n* = 17)	Malignancy (*n* = 23)	Sensitivity	Specificity	*p*-Value
At least one, *n* (%)	13 (76.5%)	23 (100%)	100%	23.5%	0.014
At least two, *n* (%)	5 (29.4%)	15 (65.2%)	65.2%	70.6%	0.025
Three, *n* (%)	1 (5.9%)	10 (43.5%)	43.5%	94.1%	0.008

* Predict factors: hypoechoic pattern, pancreatic duct dilatation, and distal pancreatic atrophy.

**Table 4 diagnostics-12-01004-t004:** Comparison factors of 23 patients with malignancy between positive and false-negative FNA/FNB cytopathological results.

	Positive (*n* = 14)	False-Negative (*n* = 9)	*p*-Value
FNA/FNB, *n* (%)	8 (57.1%)/6(42.9%)	8 (88.9%)/1(11.1%)	0.106
Pass number, *n*, mean + SD	3.43 ± 1.56	3.89 ± 1.17	0.456
Tumor location *, *n*	1/7/3/3	1/6/2/0	0.513
Tumor size, cm, mean ± SD	4.2 ± 1.26	1.98 ± 0.84	0.00
Calcification in pancreas, *n*	2 (14.3%)	2 (22.2%)	0.643

* Tumor locations: uncinate process, head, body, tail portion.

## Data Availability

Not applicable.

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
