# Peer review of "Predicting Factors for Pancreatic Malignancy with Computed Tomography and Endoscopic Ultrasonography in Chronic Pancreatitis"

_diagnostics, 2022, doi:10.3390/diagnostics12041004_

Round 1

Reviewer 1 Report

The manuscript entitled "Predicting factors for pancreatic malignancy with computed tomography and endoscopic ultrasonography in chronic pancreatitis" by Lai et al. addresses a clinical important and challenging issue.

The authors show clearly that EUS is superior of CT scan regarding diagnosis of malignancy in patients suffering of chronic pancreatitis. The manuscript is interesting to the readers of Diagnostics.

In summary the manuscript can be recommended for publication. However, it would be helpful, if the authors could address the following questions:

  • How experienced the radiologists have been in diagnosis of malignancy in CP patients?
  • Which consequences have you drawn regarding diagnostic pathways in your hospital?

Author Response

Dear Editors of “Diagnostics”,

   On behalf of all authors, I appreciate the time and effort of the editors and reviewers in critiquing our work (Manuscript ID: diagnostics-1656031), attached is a point-by-point response to reviewer comments. This manuscript has been modified according to the reviewer’s comments and thank you for consideration of publishing our work. We look forward to a favorable response and please feel free to contact me should you have any question.

Yours sincerely,

Dr. Ching-Chung Lin

Department of Internal Medicine, Mackay Memorial Hospital, Taipei, Taiwan

Reviewer 2 Report

This is clean research, but the retrospective design and the small amount of patients is an impingement for scientific soundness. The conclusions were expected, the discussions should have addressed also the MRCP as a diagnostic tool, and the not-so-detailed discussions give an overall sentiment that the article was written in a hurry.

Author Response

(The authors gave the same response as above.)

Round 2

Reviewer 2 Report

Thank you! I will leave the decision to the editors, feeling that a spectacular progress cannot be made. Good luck!

This manuscript is a resubmission of an earlier submission. The following is a list of the peer review reports and author responses from that submission.